# PeerJ

# GroopM: an automated tool for the recovery of population genomes from related metagenomes

Michael Imelfort[1], Donovan Parks[1], Ben J. Woodcroft[1], Paul Dennis[1], Philip Hugenholtz[1] and Gene W. Tyson[1,2]

[1] Australian Center for Ecogenomics (ACE), School of Chemistry and Molecular Biosciences, The University of Queensland, St Lucia, QLD, Australia
[2] Advanced Water Management Center (AWMC), The University of Queensland, St Lucia, QLD, Australia

## ABSTRACT

Metagenomic binning methods that leverage differential population abundances in microbial communities (differential coverage) are emerging as a complementary approach to conventional composition-based binning. Here we introduce GroopM, an automated binning tool that primarily uses differential coverage to obtain high fidelity population genomes from related metagenomes. We demonstrate the effectiveness of GroopM using synthetic and real-world metagenomes, and show that GroopM produces results comparable with more time consuming, labor-intensive methods.

## INTRODUCTION

Our ability to understand the function and evolution of microbial communities has been hampered by an inability to culture most component species in any given ecosystem (*Hugenholtz, Goebel & Pace, 1998*). Metagenomics, the application of shotgun sequencing to environmental DNA, has provided a means to bypass this cultivation bottleneck and obtain genomic data broadly representative of a microbial community (*Handelsman, 2004*). Historically it has not been possible to assemble the genomes of component species from complex communities due to insufficient sequence coverage, therefore tool development has largely focused on classification algorithms that assign taxonomy to sequence fragments (including reads) based on sequence composition (*McHardy et al., 2006*), homology (*Huson et al., 2007*), phylogenetic affiliation (*Krause et al., 2008*; *Wu & Eisen, 2008*) or a combination of these approaches (*Brady & Salzberg, 2009*; *MacDonald, DH & Beiko, 2012*; *Parks, MacDonald & Beiko, 2011*). The main limitation underlying these methods is their reliance on reference databases with a skewed genomic representation of microbial diversity (*Hugenholtz, 2002*).

Recent technological advances have permitted cost-effective deep sequencing (>50 Gbp) of metagenomes providing the resolution necessary to obtain partial or near complete genomes from rare populations (<1%) in complex communities

Corresponding authors
Michael Imelfort,
michael.imelfort@gmail.com
Gene W. Tyson, g.tyson@uq.edu.au

(*Albertsen et al., 2013*; *Wrighton et al., 2012*). However, the task of binning anonymous assembled sequence fragments (contigs) from a metagenome into groups representative of their source populations remains a significant challenge. One approach for binning deep metagenomes has been to cluster contigs with similar tetranucleotide frequencies using emergent self organizing maps (TF-ESOMs) (*Wrighton et al., 2012*) or interpolated markov models (*Strous et al., 2012*). A limitation of composition based methods is low binning accuracy of short contigs (<2 Kbp) and contigs from related microorganisms with similar tetranucleotide frequencies (*Teeling et al., 2004a*; *Teeling et al., 2004b*).

New binning approaches have recently emerged that primarily use differential coverage patterns (coverage profiles) across multiple related metagenomes (*Albertsen et al., 2013*; *Alneberg et al., 2013*; *Karlsson et al., 2013*; *Nielsen et al., 2014*; *Sharon et al., 2013*). These approaches are based on the assumption that contigs with similar coverage profiles are likely to have originated from the same microbial population and show that combining coverage profiles with composition based approaches holds great promise for improving binning fidelity. Here we present GroopM, a tool that automatically bins contigs into discrete population genomes based primarily on co-varying coverage profiles across multiple related metagenomes, for example temporal or spatial series of a given ecosystem.

## MATERIALS AND METHODS

### Overview of the GroopM algorithm

The coverage profiles used by GroopM are generated by mapping reads from each related metagenome onto an assembly of all or part of the same data. Contigs generated by any of the many popular short read assemblers (e.g., Velvet (*Zerbino & Birney, 2008*), SPAdes (*Bankevich et al., 2012*), IDBA-UD (*Peng et al., 2012*), Ray Meta (*Boisvert et al., 2012*)) can be used as input for GroopM binning. A typical GroopM workflow progresses through five stages: parse, core, refine, recruit and extract (Fig. 1). In the *parse* stage, GroopM loads and stores the data and creates the various profiles used in the other stages (e.g., coverage profiles). The second stage, *core*, produces a collection of preliminary bins which can be optionally refined and expanded in the *refine* and *recruit* stages respectively. Finally, the *extract* stage includes a number of printing and data extraction tools, which can be used to move GroopM specific data (e.g., bin assignments) into standard file formats (e.g., FASTA, csv) for use by other programs in downstream analyses. A detailed description of each of these stages is provided in the supplementary methods. GroopM also provides a number of plotting options that allow users to visualize the layout of their assembly in the various profile spaces (e.g., coverage profile, tetranucleotide frequency). GroopM is licensed under the GPL version 3 and is freely available at www.github.com/minillinim/GroopM.

### GroopM validation: comparison with the TF-ESOM method using synthetic data

#### Simulating metagenomic reads

The relative abundances of community members in the synthetic data used in this comparison was modeled on microbial community profiles (16S rRNA gene 97%

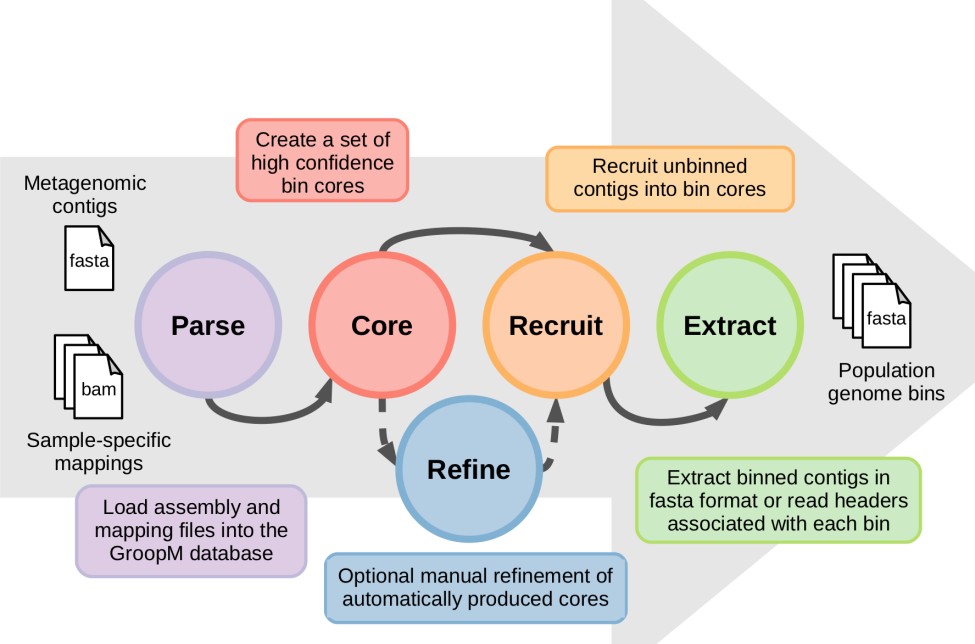

**Figure 1  An overview of the GroopM workflow.** GroopM has five stages, beginning with file parsing and ending with bin extraction. The refine step is optional and can be carried out at any stage after "core" has completed.

operational taxonomic unit [OTU] tables) from three related soil samples from Stordalen mire (*Mondav et al., 2014*). The OTU table generated from these samples contained 1,159 unique OTUs. Error-free 100 bp paired-end reads were generated from fully sequenced and permanent draft genomes (reference sequences) which were downloaded from the Integrated Microbial Genomes (IMG) (http://img.jgi.doe.gov) database, version 4.0 (see Note S1).

The metagenomic_mincer package (https://github.com/wwood/metagenomic_mincer) was used to filter this collection of references and to create the synthetic reads. Oversampling of lineages that have been the subject of concentrated sequencing efforts was avoided using the remove_strains script in this package. The script detects when strains are from the same species by searching for perfect matches in species name fields in the IMG metadata. Note that this filtering strips most strain heterogeneity from the synthetic data resulting in metagenomes with substantially simplified microbial community diversity and resulting assemblies that include fewer and longer contigs than would be expected for assemblies of a typical soil metagenome. The mince script was used to match reference sequences to OTUs and their corresponding abundance levels. For each OTU present in the table, a bacterial or archaeal genome was chosen at random from the filtered IMG reference set and synthetic reads were generated in accordance with the corresponding sample relative abundance. Reads were generated using sammy (https://github.com/minillinim/sammy) through the sammy_runner script in metagenomic_mincer. In total, 128,229,878, 121,497,178 and

114,931,846 reads were generated for each sample respectively (Data S1). Data is available at https://github.com/minillinim/GroopM_test_data/.

### Assembly of synthetic reads using Velvet

The synthetic reads were co-assembled using Velvet version 1.2.07 (*Zerbino & Birney, 2008*) with a kmer size of 85 bp and an estimated insert size of 500 bp. The expected coverage and coverage cutoff parameters were determined automatically by Velvet. After assembly, all contigs shorter than 500 bp were removed leaving 5,668 contigs totaling 133 Mb, with a longest contig of 1,667,234 bp and an N50 of 122,645 bp. These contigs are referred to below as *filtered contigs*.

### Defining verified bin assignments

The verified bin assignments for each filtered contig were determined by alignment to the 1,159 IMG reference genomes used to generate the data using BLAST version 2.2.25+ (*Camacho et al., 2009*). Almost all (>99%) of the filtered contigs aligned to their corresponding reference sequences with no mismatches. A total of 305 reference genomes were identified as the best match for at least one filtered contig indicating that perfect binning of the filtered contigs should produce 305 genome bins. The largest number of contigs mapped to a single reference was 696 for *Emticicia oligotrophica* GPTSA100. Only 69 reference genomes were the best match for at least 10 contigs and only 71 were assembled into contigs totaling at least 25 Kbp.

### Reduction of bin counts to account for chimeric contigs

During the calculation of the verified bin assignments described above it was observed that a number of contigs mapped equally well to multiple closely related reference genomes. The reference genomes in question form two groups: the first group contains five genomes of the genus *Thermotoga*; *T. maritima* MSB8, *T. petrophila* RKU-1, *T. neapolitana* DSM 4359, *Thermotoga sp*. RQ2 and *T. naphthophila* RKU-10. The second consists of two strains of the species *Oenococcus oeni*, PSU-1 and DSM 20252. These genomes were not labeled as strains during data creation due to an error in the metadata. An analysis of read mapping data revealed that 93.1% of reads generated from the *O. oeni* PSU-1 strain mapped to the DSM 20252 strain. Similar results were found for reads generated for some of the *Thermotoga* genomes. These high percentages of shared reads indicate that contigs mapping to these reference sequences could have been chimerically constructed from reads generated from a mixture of reference sequences. Therefore, it was decided that the two *Oenococcus* and the five *Thermotoga* variants should be treated as two single populations leaving 300 *verified bins* after collapsing strains.

### Binning the filtered contigs using the TF-ESOM approach

TF-ESOM binning was performed using Databionics ESOM tools (http://databionic-esom.sourceforge.net) as described previously (*Wrighton et al., 2012*). Briefly, the frequencies of all possible tetranucleotide sequences were calculated using a custom Perl script with a 1 bp sliding window that summed pairs of reverse complementary tetranucleotides. Contigs longer than 5 Kbp were split into 5 Kbp fragments and contigs

shorter than 2 Kbp were excluded. Tetranucleotide frequencies were normalized by contig length and application of the 'Robust ZT' transformation built into ESOM tools. TF-ESOMs were toroidal and used Euclidean grid distances and dimensions scaled from the default map size ($50 \times 82$) as a function of the number of data points, to a ratio of approximately 5.5 map nodes per data point. The Batch algorithm ($k = 0.15\%$, 20 epochs) was used for training and the standard best match search method was used with local best match search radius of 8. Remaining training parameters were as follows: Euclidean data space function; starting value for training radius of 50 with linear cooling to 1; Gaussian weight initialization method; starting value for learning rate of 0.5 with linear cooling to 0.1; Gaussian kernel function. Data points were assigned to classification groups (genome bins) by manually identifying the boundaries that were apparent using a distance-based background topology (U-Matrix) representation of the TF-ESOM. Data points between bins or on borders were not assigned to bins (Fig. S1B).

### Binning the filtered contigs using GroopM

Synthetic reads were mapped onto the filtered contigs using BWA version 0.6.2-r126 (*Li & Durbin, 2009*) with default settings to produce three BAM files. The resulting BAM files were sorted and indexed using samtools version 1.8 (*Li et al., 2009*). The filtered contigs and corresponding BAM files were parsed into GroopM version 0.2.0 and coring was carried out for all contigs that were at least 1 Kbp long, resulting in the formation of 63 core bins. Binned contigs were saved to disk using GroopM extract and evaluated for contamination and completeness based on the presence of 111 single copy marker genes (*Dupont et al., 2011*) using CheckM (http://ecogenomics.github.io/CheckM/). The results of this analysis were used during the running of GroopM refine to guide splitting and merging operations. First, bins with the lowest completion scores were examined using GroopM's visualization tools to see if any mergers were possible, resulting in 12 merge operations and one subsequent split operation. Next, bins with the highest contamination scores were examined using GroopM's visualization tools to determine if they were chimeras, resulting in three split operations (one which divided a bin into three parts), one merge operation and two bin deletions. These operations reduced the number of bins to 53. The resulting genome bins were expanded to include contigs <1 Kbp using GroopM recruit. Binned contigs were saved to disk using GroopM extract and evaluated again for contamination and completeness.

### Evaluating binning accuracy

Bins obtained using both methods were compared with the set of 300 verified bins described above using the following method: first, each bin was assigned to the verified bin that contained the majority of its contigs. Whenever two or more bins could be assigned to the same verified bin, the competing bins were ranked in descending order of the cumulative length of contigs belonging to the verified bin and only the top ranked bin was assigned to the verified bin. All other bins were assigned to the verified bin that the next largest number of their contigs belonged to. This procedure was applied iteratively until such time that each TF-ESOM and GroopM bin was coupled with exactly one verified

bin and that no verified bin was coupled with more than one TF-ESOM or GroopM bin. Contigs were then classified as assigned correctly, incorrectly or not assigned to any bin. Binned contigs that belonged to the same verified bin that their respective bin was coupled with were classified as binned correctly. All other binned contigs were classified as binned incorrectly. All bins were evaluated for contamination and completeness based on the presence of 111 single copy marker genes (*Dupont et al., 2011*) using CheckM (Tables S1 and S2).

## Recovery of population genomes from a human gut microbiome

### *Data preparation*

Unassembled data consisting of 18 paired Illumina short read data files sequenced from 11 fecal samples was downloaded from the NCBI Sequence Read Archive (http://www. ncbi.nlm.nih.gov/sra) on the 29th of May 2013. The corresponding binned contigs (*Sharon et al., 2013*) (Sharon assembly) were downloaded from http://ggkbase.berkeley.edu/ on the 28th of May 2013 comprising 2,998 contigs separated into 33 groups, two of which appear to be associated with contigs which could not be classified (CARUNCL and ACDRDN).

The published assembly process contained a number of tasks that would normally be categorized as binning, comprising numerous iterations including a number of steps that focused on identifying and obtaining genomes that differed at the strain level. In order to provide an unbiased test of GroopM using strictly unbinned contigs, the raw data was re-assembled prior to binning. Prior to assembly all reads were hard trimmed to 85 bp. Reads containing any sequence beginning with three or more bases with quality score 2 were trimmed from the start of this sequence until the end of the read. Finally, all reads shorter than 50 bp were removed along with their corresponding paired read.

### *Assembly and binning of the Sharon dataset using SPAdes and GroopM*

The trimmed reads were assembled with SPAdes 2.4.0 (*Bankevich et al., 2012*) using default parameters. As this version of SPAdes cannot handle 18 separate data files, all of the trimmed reads files were concatenated to produce a single paired read data set. The resulting assembly was filtered to remove all contigs shorter than 500 bp leaving 7,016 contigs totaling 38.7 Mb, with a longest contig of 406 Kbp and an N50 of ∼36 Kbp. Trimmed reads were mapped onto the SPAdes contigs with BWA-mem and the resulting output was parsed into GroopM using the same approaches and software versions described above for the synthetic data assembly. GroopM produced 24 core bins using all contigs that were at least 1 Kbp long. Assessment of bin quality, bin refinement and contig recruitment were performed using the method described above resulting in a single split operation, raising the total number of bins to 25. Assembled contigs are available at https://github.com/minillinim/GroopM_test_data.

### *Visualizing the Sharon assembly using GroopM*

The trimmed reads were mapped onto the Sharon assembly using BWA-mem 0.7.5a (*Li, 2013*) and the resulting 18 BAM files were sorted and indexed using samtools 1.8 (*Li et al., 2009*). The Sharon assembly and corresponding BAM files were parsed into GroopM

version 0.2.0 and the resulting GroopM database was modified to incorporate the Sharon assembly bin assignments allowing visualization using GroopM.

***Matching GroopM bins with the Sharon assembly***

Each SPAdes contig was linked to at most one of the contigs from the Sharon assembly using nucmer version 3.07 (*Delcher et al., 2002*). Reciprocal best hit linkages were computed with a custom python script (http://github.com/minillinim/taintedSwallow). A minority of SPAdes contigs could not be linked to a Sharon contig and vice versa, however, this is to be expected as the two approaches used substantially different assembly approaches.

## Assessing the effects of using different assembly algorithms on GroopM output

The trimmed Sharon data was assembled using three assemblers: SPAdes version 2.4.0 (*Bankevich et al., 2012*) (default parameters), CLC Bio version 7.0.3.64 (http://www.clcbio.com, default parameters) and Velvet version 1.2.07 (*Zerbino & Birney, 2008*) (kmer 45 bp). For each assembly, all contigs ≥500 bp were binned with GroopM using the process described above for the synthetic data. Prodigal version 2.60 (*Hyatt et al., 2010*) (default parameters + meta switch) was used to identify open reading frames (ORFs) in the Sharon bins and the three sets of GroopM bins. The Sharon ORFs were aligned to each set of GroopM ORFs using Nucmer version 3.07 (*Delcher et al., 2002*) (default parameters) and the longest reciprocal match for each Sharon ORF was determined using a custom python script (identity ≥ 99%, length ≥ 60%). The majority of Sharon ORFs from each bin mapped to a single corresponding GroopM bin (dominant bin) with the remainder mapping to several other bins. The similarity of GroopM outputs using different assemblers was assessed using all Sharon bins with at least 1,000 ORFs (11 bins).

## RESULTS AND DISCUSSION

We validated GroopM using synthetic metagenomes constructed from 1,159 reference genomes with coverage patterns modeled on three related soil habitats. Co-assembly of these synthetic metagenomes produced 5,668 contigs (>0.5 Kbp), however, most genomes were present at low coverage and did not assemble. Contigs were derived from 305 reference genomes (300 verified bins), 71 of which each had a combined contig length of at least 25 Kbp (Data S1). We compared bin assignments made using GroopM and TF-ESOM. Binning accuracy was assessed based on agreement with verified bin assignments taking into account both total numbers of contigs (TNC) and total assembled bases (TAB) that were assigned correctly, incorrectly or not assigned to any bin. GroopM correctly binned 79.6% of the synthetic contigs (97.5% TAB), and performed substantially better than TF-ESOM (45.9% TNC and 84.2% TAB). A much larger proportion of the metagenome was correctly binned with GroopM (∼73% more contigs than TF-ESOM), and only 4.7% of the contigs were incorrectly binned (0.4% TAB) as opposed to 5.7% (8.7% TAB) for TF-ESOM. Furthermore, the GroopM errors were primarily restricted to short contigs (0.5 to 2 Kbp) with only 62 contigs longer than 2 Kbp incorrectly assigned,

including nine contigs between 10 and 30 Kbp. In contrast, TF-ESOM incorrectly assigned 193 contigs longer than 5 Kbp, including 31 contigs longer than 100 Kbp (Table S3).

We analyzed binning errors using visualizations of contig distributions and bin assignments in coverage profile and tetranucleotide frequency spaces (Fig. 2). Differential coverage and tetranucleotide frequency binning approaches may perform poorly in the presence of different populations with highly similar coverage profiles and tetranucleotide frequencies, respectively. As predicted, the majority of the GroopM errors (62.4%) include populations with highly similar coverage profiles (Figs. 2B, 2D and 2F), while most TF-ESOM errors (81.3%) cluster around populations with closely matching tetranucleotide frequencies (Figs. 2A, 2C and 2E). The majority of GroopM errors were localized to seven genome bins (Note S2), while the TF-ESOM errors appeared more systemic, affecting 61 genome bins. The TF-ESOM errors were mostly due to difficulties in identifying boundaries in ESOMs that separate contigs with highly similar tetranucleotide frequency profiles (Note S3). Our results show that population genome bins can be more accurately resolved using coverage profiles augmented by tetranucleotide frequencies than by using tetranucleotide frequencies alone (Fig. 3, Note S4).

We further validated GroopM using 18 metagenomic datasets sourced from an infant human gut microbiome previously analyzed by *Sharon et al. (2013)* using a custom iterative assembly and binning process that included a coverage-based ESOM binning step. This produced 2,998 contigs (>407 bp) partitioned into 33 bins comprising microbial, viral and plasmid genomes. To provide an unbiased test of GroopM, we reassembled the raw sequence data from the 18 datasets using SPAdes (*Bankevich et al., 2012*) with default parameters producing 7,154 contigs (>500 bp). GroopM clustered 6,959 of these contigs (99.5% TAB) into 25 population genome bins using default parameters and minimal refinement steps. The completeness and contamination of the Sharon and GroopM bins were estimated based on the presence and copy number of 111 conserved single-copy bacterial marker genes (*Dupont et al., 2011*) using CheckM (Tables S4 and S5).

Overall, the two approaches produced highly similar population genome bins (Fig. 4). The six most complete GroopM bins corresponded to the seven most complete Sharon bins, with all bins having estimated completeness of greater than 70% (Tables S4 and S5). The additional Sharon bin is one of two dominant *Staphylococcus epidermidis* strains that were co-assembled and combined by GroopM into a single bin (GM_76). While GroopM did not reconcile individual *S. epidermidis* strains, it grouped all of the contigs associated with the strains together into a small number of bins suitable for manual refinement (Note S5). GroopM produced a number of genome bins which were more complete than those originally reported, including two *Propionibacterium* population genomes: GM_11 (2.76 Mb, 89.2% complete) and GM_19 (1.1 Mb, 13.5% complete) that correspond to Sharon bins CARPRO (2.52 Mb, 88.3% complete) and CARPAC (0.36 Mb, 2.7% complete) respectively (Fig. S2 and Note S6). The most striking example of genome recovery differences was the complete absence of a partial GroopM bin (GM_34, 73 contigs, 75.6 Kbp, 39% GC) in the original study that was not detected in their contigs (Figs. S3 and S4). Analysis of alignments of contigs from this bin to IMG reference genomes

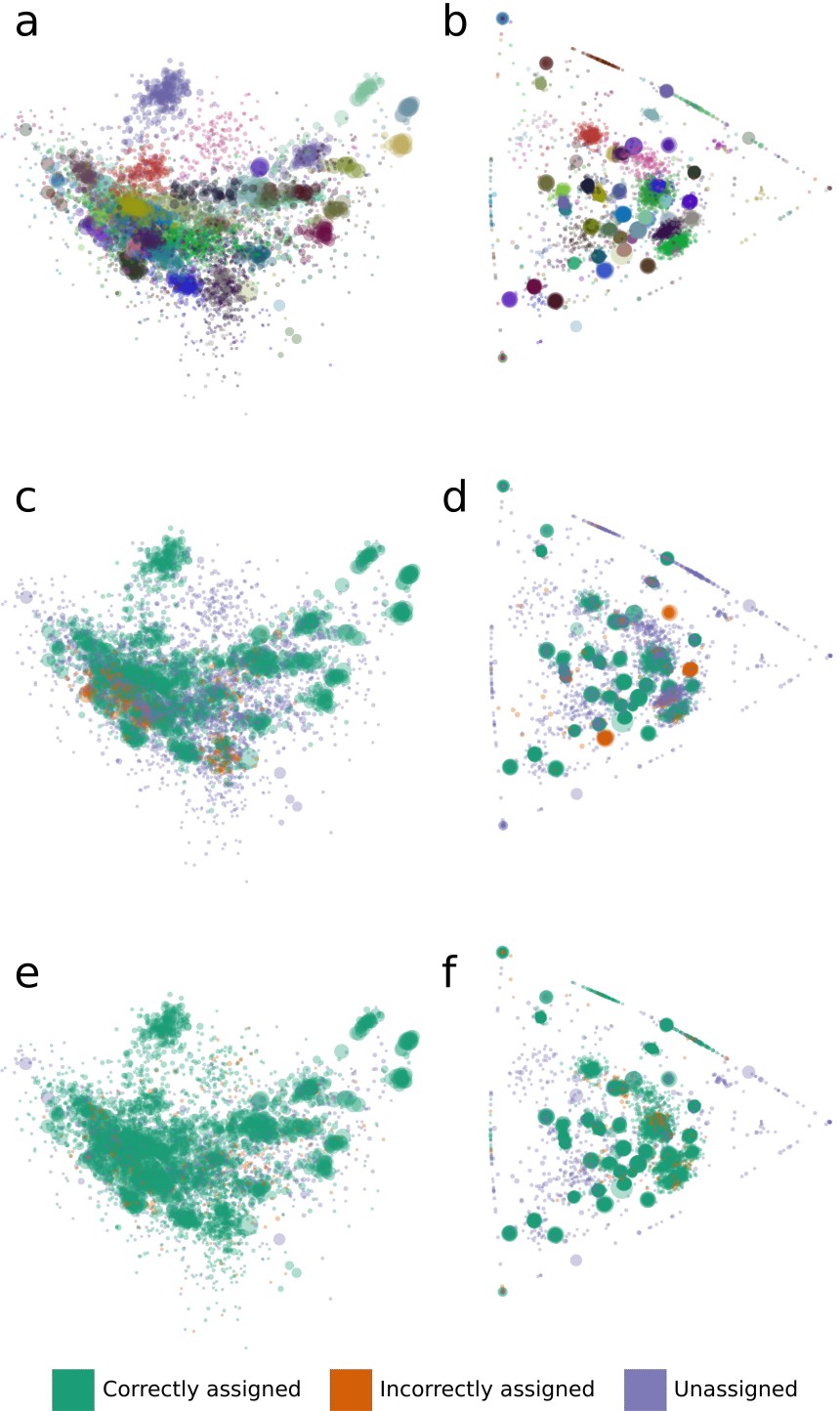

Correctly assigned    Incorrectly assigned    Unassigned

**Figure 2  The distribution of tetranucleotide frequencies, coverage profiles and bin assignments for the synthetic metagenomic contigs.** The diameter of each circle is proportional to the length its respective contig. (A, C, E) Contigs are positioned according to the first two principal components of their tetranucleotide frequencies. The first principal component is positioned 

**Figure 2 (...continued)**

horizontally, the second is positioned vertically. (B, D, F) Contigs are positioned according to their $x$ and $y$ coordinates in GroopM transformed coverage profile space. (A, B) Each 'true' bin is assigned a random color and contigs are colored according to their true bin assignments. (C, D) Contigs are colored according to the accuracy of their bin assignments using TF-ESOM. (E, F) Contigs are colored according to the accuracy of their bin assignments using GroopM.

indicates that GM_34 is most likely a member of the *Proteobacteria* (Data S1). The GM_34 population appears to spike in abundance at day 21, but is present at relatively low levels in all the other samples. It is possible that the lack of good representation in multiple samples may have resulted in its poor assembly. We speculate that differences in genome bin recovery are due to the iterative assembly method used to create the Sharon contigs which may have filtered reads from some genome bins resulting in partial or complete absences in the final assembly. Therefore caution should be exercised when using methods that involve strict filtering of the data, especially if the goal of the analysis is to infer that the resulting population bins or microbial communities are missing key metabolic components. Despite these and a small number of other minor binning differences (Tables S4 and S5), GroopM was able to produce highly similar population bins to *Sharon et al. (2013)* in less than 2 h with a small memory footprint (Table S6). Moreover, the GroopM binning was based on a single assembly of the metagenomic data using default parameters, as opposed to the numerous assemblies and filtering steps described in the original study.

We established that GroopM is not substantially affected by choice of assembler by repeating the binning on two additional generic assemblies using Velvet and CLC and analyzing open reading frame (ORF) distributions in the Sharon and GroopM bins. The CLC Bio and SPAdes assemblies were very similar (SPAdes: 7,016 contigs, total ~38.7 Mbp, N50 35,986 bp, CLC: 7,576 contigs, total ~37.7 Mbp, N50 29,979 bp) (Fig. S5), however the Velvet assembly was sub-optimal (7,199 contigs, total ~29.7 Mbp, N50 8,272 bp) and only included ~75% of the bases included in the Sharon, CLC Bio and SPAdes assemblies. Interestingly, the missing data from the Velvet assembly originated from a small number of microorganisms whose contigs were almost completely missing from the assembly. Of the contigs that Velvet did produce, the binning was consistent with the CLC Bio and SPAdes-derived bins. We hypothesize that this could be the result of Velvet's reliance on coverage cutoffs and highlights the motivation behind the multiple assembly and filtering steps used to create the assembly published by *Sharon et al. (2013)*. The consistency of binning outcomes indicates that GroopM's output is not markedly affected by the choice of assembly algorithm used, provided that the chosen assembler produces an "adequate" assembly. The low proportion of matching ORFs for the CARSEP and CARSEP3 bins is the result of GroopM's inability to reliably resolve bins at the strain level (Note S2). GroopM consistently groups most of the contigs from CARSEP and CARSEP3 together into a single bin whose ORFs are almost evenly divided between the two Sharon bins. The similarity of same strain groupings for all three assemblies provides further evidence that GroopM's output is largely independent of the choice of assembler used.

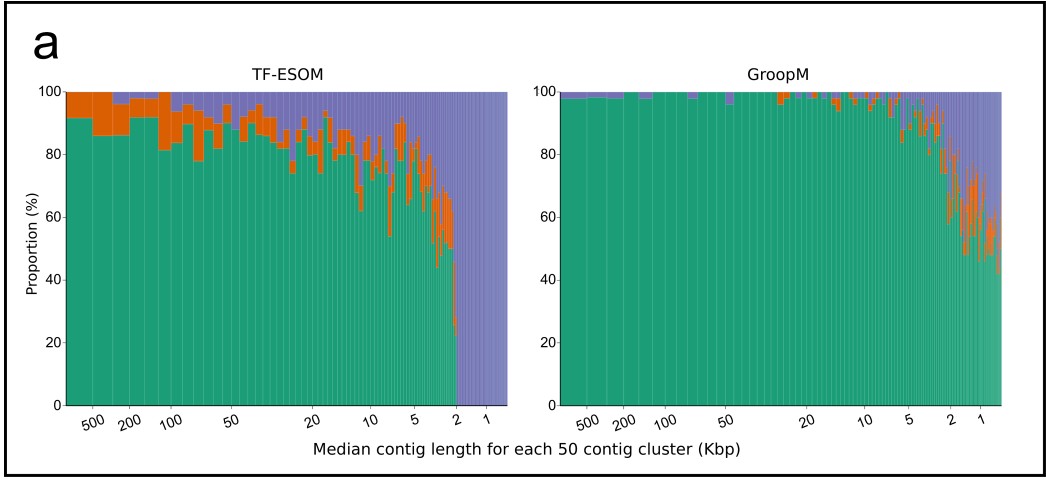

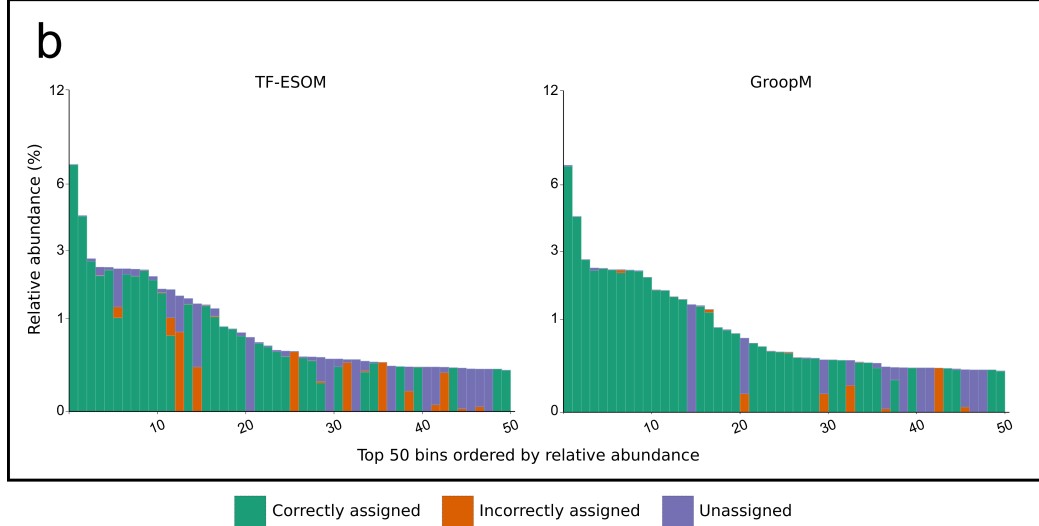

**Figure 3** **An overview of the relationships between contig length, population relative abundance and binning accuracy for the TF-ESOM and GroopM approaches.** (A) Contigs are ordered from longest to shortest and grouped together into clusters of 50. Each bar represents a single cluster and has a width that is proportional to the total number of assembled bases in that cluster. Bars are split vertically according to the percentage of their bases that are either correctly, incorrectly or not assigned. The large region of unassigned contigs in the TF-ESOM plot reflects the lower binning limit of 2 Kbp for this method. (B) Verified bins are ordered in descending relative abundance, calculated based on the number of simulated reads created using each reference. Each bar represents a single verified bin and the height of each bar represents the bin's relative abundance. Bars are split vertically according to the percentage of their bases that are either correctly, incorrectly or not assigned by the corresponding method. Both methods had decreased accuracy for very low abundance bins however GroopM was able to correctly bin nearly all the contigs for the most dominant species.

## CONCLUSIONS

In summary, GroopM automates production of high fidelity population genome bins from related metagenomes by primarily leveraging differential coverage profiles. Decreasing sequencing costs and the push for increased replication (*Prosser, 2010*) will make this a feasible approach for many metagenomic projects. GroopM can bin contigs that have been
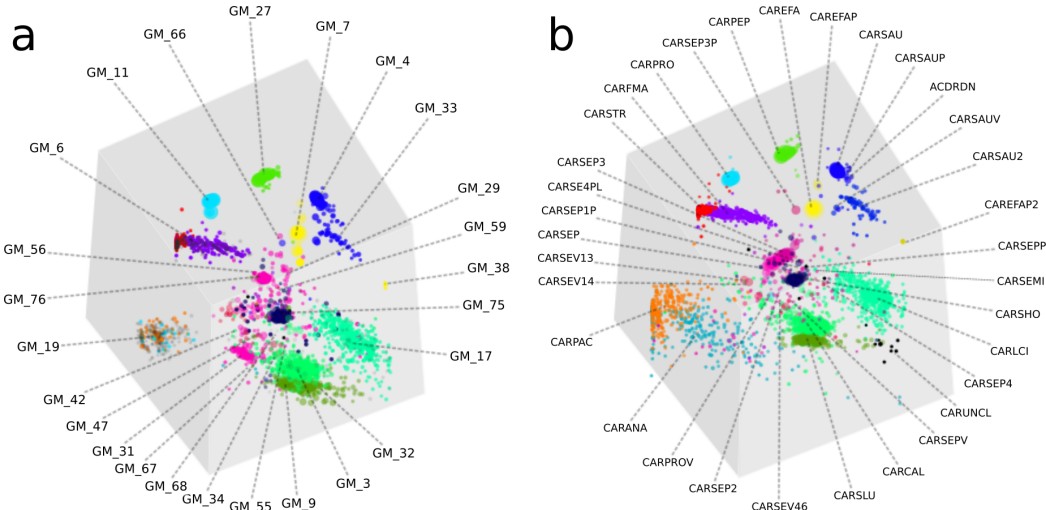

**Figure 4 A comparison of GroopM and Sharon bin assignments generated using visualization tools within GroopM.** (A) Contigs and resulting bins made using SPAdes and GroopM. (B) The Sharon assembly visualized in GroopM coverage space. All the contigs belonging to a single bin are assigned the same color. Each bin was assigned a random unique color with the exception of strain variants which were assigned very similar colors. GroopM-binned contigs are colored according to the bin assignment of their closest matching contig in the Sharon assembly.

generated using a range of assembly methods including co-assemblies and single sample assemblies, provided that metagenomic data is available for at least three related samples (Note S7). GroopM also provides a dedicated tool for visual interactive metagenomic bin editing. The software allows users to view and merge bins, as well as split bins based on composition, coverage or contig length profiles. Resolving population bins of closely related co-habiting genotypes remains a challenge for all binning approaches. The current implementation of GroopM places contigs from closely related genotypes into chimeric bins, as seen in the *Sharon et al. (2013)* dataset (Note S5), which require manual curation to separate.

During the review of this manuscript, we became aware of two studies that use similar coverage-based approaches to bin population genomes from related metagenomic datasets. *Nielsen et al. (2014)* apply the method at the gene level, retrieving sets of genes that share co-abundance profiles and *Alneberg et al. (2013)* report a tool called CONCOCT that combines differential coverage and tetranucleotide frequency profiles with linking reads to bin contigs into population genomes. It is beyond the scope of the present study to compare GroopM to these methods, but assessing their relative strengths and weaknesses is clearly an important objective for future studies.

GroopM is currently being used to recover hundreds of high fidelity population genomes from numerous habitats including anaerobic digesters, permafrost and host-associated systems (human, coral, insect, plant). Such genomes serve two important purposes. Firstly, they rapidly fill out the microbial tree of life, in particular by providing the first genomic representation for many candidate phyla, which is important for correcting our currently skewed understanding of microbial evolution. Secondly, they

provide a basis for development of genome-based trophic interaction networks to facilitate understanding of how microbial communities function in given ecosystems. These developments will ensure that microbiologists can make best use of the opportunities presented by the ongoing high throughput sequencing revolution.

## ACKNOWLEDGEMENTS

We thank Harald Gruber-Vodicka and an anonymous reviewer for their useful input.

### Funding

This study was supported by the Commonwealth Scientific & Industrial Research Organisation (CSIRO) Flagship Cluster "Biotechnological solutions to Australia's transport, energy and greenhouse gas challenges". PH was supported by a Discovery Outstanding Researcher Award (DORA) from the Australian Research Council (DP120103498). GWT was supported by an ARC Queen Elizabeth II fellowship (DP1093175). The funders had no role in study design, data collection and analysis, decision to publish, or preparation of the manuscript.

### Grant Disclosures

The following grant information was disclosed by the authors:
Commonwealth Scientific & Industrial Research Organisation (CSIRO) Flagship Cluster "Biotechnological solutions to Australia's transport, energy and greenhouse gas challenges".
Australian Research Council (ARC) Discovery Outstanding Researcher Award (DORA): DP120103498.
ARC Queen Elizabeth II fellowship: DP1093175.

### Competing Interests

The authors declare there are no competing interests.

### Author Contributions

- Michael Imelfort conceived and designed the experiments, performed the experiments, analyzed the data, contributed reagents/materials/analysis tools, wrote the paper, prepared figures and/or tables, reviewed drafts of the paper.
- Donovan Parks and Ben J. Woodcroft conceived and designed the experiments, performed the experiments, contributed reagents/materials/analysis tools, reviewed drafts of the paper.
- Paul Dennis analyzed the data, reviewed drafts of the paper.
- Philip Hugenholtz and Gene W. Tyson conceived and designed the experiments, wrote the paper, prepared figures and/or tables, reviewed drafts of the paper.

## Supplemental Information

Supplemental information for this article can be found online at http://dx.doi.org/10.7717/peerj.603#supplemental-information.

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
