# Peer review of "GroopM: an automated tool for the recovery of population genomes from related metagenomes"

_PeerJ, doi:10.7717/peerj.603_

## Round 0.1 · original submission · Minor Revisions

I think a comprehensive test against competing tools is too much to ask for the authors, however a brief mention of these already published tools is probably a good idea.

Reviewer 1 ·

Basic reporting

The authors present the novel method "GroopM" for binning metagenomic datasets into taxon-specific bins after a de-novo assembly. The main idea is the utilization of coverage information for the contigs, therefore a number of related samples is necessary. The effectiveness of the approach if demonstrated using synthetic and real-world metagenomes.

- The authors do not mention other tools tackling the same problem, e.g. CONCOCT (http://arxiv.org/abs/1312.4038). A benchmark against CONCOCT would be nice. A second method was published while the GroopM article was under peer review, doi:10.1038/nbt.2939

- Line 361: The application scenario is interesting, but would serve better as a motivation rather than a conclusion. The fact that the authors use GroopM for a variety of other projects is nice, but doesn't really belong here.


- Supplementary Note 5 should be presented in the main manuscript, as this is the major shortcoming of their method, and users should be perfectly aware of this issue.

Experimental design

We propose to move the detailed description of the GroopM algorithm from the supplementary methods into the main manuscript. It is of general interest how the algorithm works, not only which steps it performs (this sequence of steps also is covered by the manual)

Line 60: It is not clear why Velvet and SPAdes are referenced here instead of commonly used metagenome assemblers, e.g. IDBA-UD (doi:10.1093/bioinformatics/bts174), Ray Meta (doi:10.1186/gb-2012-13-12-r122) or Omega (doi:10.1093/bioinformatics/btu395). This becomes apparent later on, when the authors mention that their benchmarking assemblies were produced with those two assemblers. But why?

Validity of the findings

- Software crashes on our real-world data set. This error needs to be fixed (or explained, if it is due to the nature of our data set). We are willing to continue working with the authors to identify the problem.
- The synthetic dataset the authors provided does work on our system, if we only run the GroopM part of the analysis, using the contigs and BAM files provided. However, we were not able to reproduce the complete pipeline, as assembly parameters and
mapping parameters (and which tools were used) were not provided.

Additional comments

We wish to highlight the authors' GitHub page (http://minillinim.github.io/GroopM/), providing a brief overview, installation instructions, an (incomplete, but still useful) user manual and the API documentation. This serves well as a starting point for end users. Also, all custom scripts are made available on GitHub as far as we can see, so (in theory, we did not try) all results can easily be reproduced.


* Line 102: Velvet version needs to be 1.2.07, not 12.07

·

Basic reporting

Missing prior literature:
L344-346 describe groopm as the first software to perform visual interactive metagenomic bin editing, but they read like the description of metawatt, a interactive gui based binning tool that was published in Strous et. al in 2012, has been constantly developed and is available at http://sourceforge.net/projects/metawatt/. Metawatt also facilitates the binning process by providing taxonomic as well as completeness information on any generated bin.
CONCOCT is a very similar tool to groopm and has been publicly described in http://arxiv.org/abs/1312.4038 and is available from https://github.com/BinPro/CONCOCT. The description from the webpages states: A program for unsupervised binning of metagenomic contigs by using nucleotide composition, coverage data in multiple samples and linkage data from paired end reads. Preliminary tests by collaborators of me show that CONCOCT performs similarly to groopm.

Experimental design

Missing controls: In software evaluation once a tools works in a general sense the important 'control' is to cross-validate its performance with other similar software tools.
I would suggest that the authors try to evaluate the performance of their semi-unsupervised binning tool against other tools of this kind such as metawatt and CONCOCT, see 'Basic reporting' section

A binning process should be evaluated in a gradient of different complexity of samples. You should test groopM with the simplest datasets possible - single/dual species setups and show how well it performs on this end of the diversity spectrum.

What about sample numbers, does that affect binning quality. I would expect that the binning accuracy is positively correlated with sample numbers, could you test for that? what about fixing the reference genomes to a realistic number (e.g. 30) and then generating more read sets with different abundances and then looking how well groopm performs with different numbers of read sets?

Missing methods and descriptions in L156-158:
Binned contigs were [...] evaluated for contamination and completeness based on the presence of 111 single copy marker genes (Dupont et al. 2011). - This essential part of the binning process is not reproducible without the scripts used. GroopM as provided on June 15 2014 (last check) provides an automated binning process for already assembled metagenomes, leaving the user with a number of bins (15 – hundreds in my tests) most of which are incomplete draft bins that need refinement, (mergings and splittings mostly as the authors also describe) to provide correct population genome level bins. To evaluate the taxonomic composition and completeness of these bins, additional steps need to be taken and the authors in L156 – 167 vaguely describe this process of evaluating bin completeness and contamination using single copy marker genes, but this essential part of the binning process that moves the draft bins to final high fidelity population genomes is completely missing from the groopm distribution that is available for download.

Validity of the findings

As it is distributed right now groopM alone can not recover high fidelity population genome bins as stated by the authors in L351-353 in the Conclusions section. groopM in its current form can provide draft bins, which can then serve as the basis to recover high fidelity population genome bins. This should be clarified.

Additional comments

As a summary, I find the software tool groopm promising, but I would suggest that the authors try to evaluate the performance of their semi-unsupervised binning tool against other tools of this kind such as metawatt and CONCOCT.
If the authors would completely provide the pipeline they use to produce high fidelity population genomes using groopm as the bin creation/editing tool the software pipeline could become a game-changer in the field of metagenomics by democratizing access to bioinformatics pipelines only accessibly to trained bioinformaticians up to now.

Minor comments:
L101 - 103: As far as I understand the combined read sets for all 3 synthetic samples were assembled together, please indicate this
L102: velvet version should read 1.2.07
L160-165 splits and merges do not add up (63-11+1+3-1=55), were more than 2 bins merged at once?
L108 – L130 ‘verified’ bin construction: I find the term ‘verified bin’ confusing, as these are the reference genomes that the reads were reconstructed from. It would have been possible to actually record the genomes the reads were generated from during the generation – e.g. in the read headers. I would suggest to rename to reference genomes, and indicate that 2 of the 300 reference genomes are actually collected genomes of closely related taxa, which is not surprising given that the bacterial taxonomy and the species naming process is sometimes problematic. What I do not understand is why one of the Oenococcus oeni strains was not removed prior to the read generation process as indicated in L85-87 (remove_strains script).
L158 Dupont et al identified 107 genes, not 111.
L156 – 167 please provide the scripts used for this process for the full functionality of groopm as a binning tool (see above for details).
170 – 183 Evaluating binning accuracy: I find the process of assigning bins to reference genomes problematic – I would assume that multiple bins can be linked to a single reference genome if e.g. the groopm but also the esom binning process was oversplitting (which I observed for groopm as well as other binning programs in my personal tests with well-defined datasets). If one then unlinks these correct assignments and forces the second best assignment just to get to a 1:1 link ratio this inflates false negative scores. I did not find numbers how often this re-assignment has happened during the analyses, this should be explored and corrected for if necessary.
L196 remove ...with a quality score...
L208: this is the paragraph for the sharon assembly reassembly, the Sharon assembly paragraph is below.
L229-230: spades assembly already described in L200-L202
L233 prodigal has a metagenomic switch where it runs in metagenomic mode and analyzes sequences even when the organism was unknown in the training stage.

Methods: In summary I found the methods part quite confusing, some obvious mistakes added to this impression. I would suggest to simplify it, focusing on the major processes:
a) read sets: 1) synthetic 2)Sharon trimmed
b) assembly processes: velvet, clc, spades
c) reference genomes selection process
d) binning 1) ESOM 2) groopm
e) bin validation synthtetic data
f) assembly method comparison

L242: The results part reads like a results and discussion, please rename accordingly
L246: 305 reference genomes but 300 verified bins according to methods, I would suggest to only use the term reference genomes and 300 as their number
257: were these the longest contigs (I assume) – then write …, the longest … instead of including
L262 why is this expected? move explanation from supplements (partially Note 4) to this paragraph, as this strengthens the groopm approach
L306-309 The filtering in assembly process could be the culprit, but this paragraph is too speculative. The ‘bin’ GM_34, 73 contigs, 75.6 Kbp is so small that it seems unreasonable to follow this methodological dissection. A functional analysis of the content of this bin might be informative though.
L315-L330 velvet is an assembler designed for bacterial genomes and not only coverage cutoffs but also wrong kmer choice and wrong expected coverage affect the quality of the velvet assembly. My personal experience with velvet in metagenomic analyses is that the default settings underestimate the expected coverage and coverage cutoff dramatically, leading to highly fragmented assemblies compared to the optimal settings for a given coverage range of a metagenomic assembly.
General comment to the assembly process: All assemblers employed are not designed for metagenome assembly. Velvet is a single bacterial genome assembler, CLC is closed source and quite expensive, only Spades with its iterative approach accross kmers can somewhat be expected to perform reasonably well, but was also designed for single genomes. I would suggest to use IDBA-UD as an assembler that was designed to work with highly uneven coverages as in metagenomes and that performs well in terms of assembly capabilities and memory usage and is freely available. Especially in the light of the proposed co-assembly of all read sets for the initial main assembly memory usage becomes an issue, but no memory usage statistics are given for any of the assemblies. This is not the main focus of the manuscript, but it could be a main culprit in the work-flow of potential groopm users.

---

## Round 0.2 · Minor Revisions

The reviewers ask for a full scale evaluation against third party tools that I believe is over the top. Please add a light weight discussion of other tools.

---

## Round 0.3 · accepted · Accept

Sorry for the confusion, there was a delay as the detailed responses were not previously made available to the reviewers (now rectified).

Reviewer 1 ·

Basic reporting

No Comments

Experimental design

No Comments

Validity of the findings

As the editor suggested, I will not ask for a comparison to similar tools again.

But: It would still be nice, though, if the authors could make their benchmark data sets
available and provide a detailed description of how to reproduce their results.
I was not able to reproduce the results they present in their manuscript.
Which assembler did they use? Which parameters? Etc.

Reproducibility is a very important aspect in research and I urge the authors to
take this seriously and document clearly what they did.


I still think that Supplementary Note 5 should be presented in the main manuscript, as this is the major shortcoming of their method, and users should be perfectly aware of this issue.

Additional comments

It would have been much easier for the re-review if you would have responded to the comments raised by the reviewers one by one. Some (minor) points both reviewers raised in the previous reviews were apparently just ignored and it would have been nice to get at least a comment why you think it does not have to be addressed.

·

Basic reporting

No Comments

Experimental design

No Comments

Validity of the findings

No Comments

Additional comments

The author’s have carefully revised their manuscript, improving on the structure and readability of the manuscript and appending informative information on the bin completeness evaluation in the supplement. Many of the two reviewer’s comments have been implemented in the revision.

As a sidenote – The changes in the manuscript were not readily track able as only a single added paragraph was highlighted as track changes when in fact quite a bit of the manuscript had been restructured, errors and omissions corrected …